

# Changes in the properties of deep and intermediate water masses in the Nordic Seas from 1997 to 2016

Małgorzata Merchel, Waldemar Walczowski

Physical Oceanography Department, Institute of Oceanology Polish Academy of Sciences, Sopot, 81-712,

Poland

*Correspondence to*: Małgorzata Merchel (merchel@iopan.gda.pl)

**Abstract.** This study investigated the temporal variability of the basic physical properties of deep and intermediate waters in the Nordic Seas from 1997 to 2016. Special attention paid to quantifying changes in their

temperature and salinity and determining potential drivers of these changes. Hydrographic data were obtained during annual cruises of the *R/V Oceania* in the Nordic Seas region from 1997 to 2016.

The results show that, in the past 20 years, deep and intermediate waters in the studied region have warmed at least 10 times more than the mean global ocean temperature change; salinity has also changed. This means that changes in these waters' properties have a much greater impact on recent climate change

intensification than previously thought. For example, ocean circulation, dissolved oxygen, carbon dioxide content, and sea level rise, may also change much faster.

1.    **Introduction**

The Nordic Seas (the Greenland, Norwegian, and Iceland Seas) (Fig. 1) are a primary location for high-latitude water mass transformation, where strong vertical mixing induced by heat loss to the atmosphere converts most of the incoming subtropical warm and saline Atlantic water into dense overflow water (Latarius and Quadfasel, 2016).




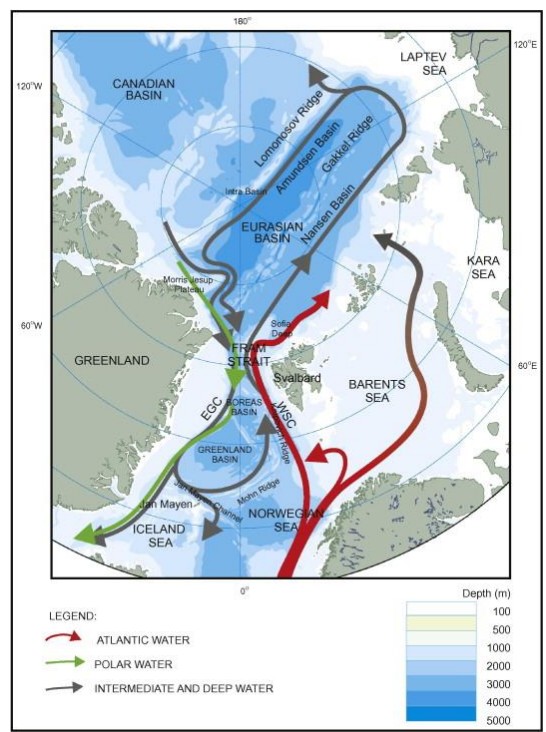

**Figure 1: Map of the Nordic Seas and the European Arctic with bathymetry and main circulation patterns, including the East Greenland Current (EGC) and West Spitsbergen Current (WSC) (Langehaug and Falck, 2012)**

The Nordic Seas and European Arctic regions have been studied by the Institute of Oceanology, Polish Academy of Sciences (IO PAN) since the early 1990s (Walczowski, 2014). Previous work was focused mainly on warm salty waters of Atlantic origin, their importance for climate (Walczowski and Piechura, 2011; Walczowski et al., 2012) and their impact on sea ice (Piechura and Walczowski, 2009). During more than 20

10    years of regular measurements, IO PAN has collected a unique set of oceanographic data.

In recent years, the attention of oceanographers and climatologists has been increasingly focused on ocean warming and the ocean's importance as a buffer that stores excess energy reaching the Earth via the greenhouse effect. Recent studies have shown that heat accumulates not only in the ocean's surface layer but also in intermediate and deep waters.

15    The warming of the ocean layer between 700 and 2000 m from 1957 to 2009 accounted for ~30% of the total global growth in the ocean's heat content (from 0 to 2000 m) (IPCC, 2013) (Fig. 2). According to Levitus et al. (2012), from 2003 to 2010, the heat content in the ocean layer from 0 to 700 m increased more slowly compared to the previous decade, and the absorption of heat at depths between 700 and 2000 m did not weaken during this period (Levitus et al., 2012). This suggests that the ocean's deeper layers contribute to the weakening

20    of surface layer warming by absorbing the latter's accumulated excess heat.





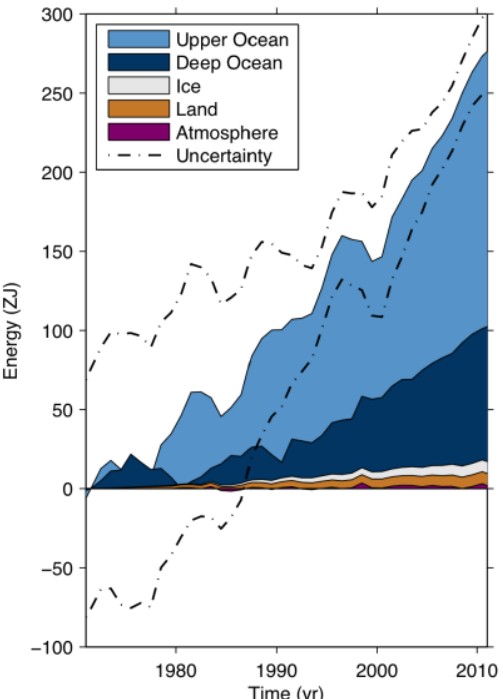

**Figure 2: Energy accumulation within distinct components of the Earth's climate system**

**(IPCC, 2013)**

Ocean coverage by hydrographic measurements below 700 m is very poor; below 2000 m, these measurements are limited solely to specific, repeated transects (IPCC, 2013). This is particularly true for the Arctic region, where harsh weather and changing ice conditions severely limit oceanographic measurements and thus data availability. Therefore, it is difficult to accurately estimate changes occurring in deep and intermediate water, while results are affected by high measurement uncertainty.

According to the IPCC, observational data sufficient to assess changes in ocean temperature below 2000 m exist from 1992 to 2005 only. According to estimates by the IPCC's Fifth Assessment Report (2013), during this period there was no significant trend in temperature changes between 2000 and 3000 m, while ocean waters warmed at depths below 3000 m, with the strongest warming observed in the Southern Ocean.

A better understanding of temporal changes in intermediate and deep water is made critical by the
important role these layers play in the global climate system. Cold, deep water is a major reservoir of dissolved carbon dioxide; its formation processes are very important for global thermohaline circulation forcing. Thermohaline circulation plays two very important roles: regulating climate by distributing heat globally (Rahmstorf, 2002) and supporting marine life by providing well-oxygenated and nutrient-rich water.

Increasing concentrations of greenhouse gases in the atmosphere warm the Earth's climate system.
More than 90% of the excess heat remaining in this system as a result of the greenhouse effect is absorbed by the oceans. The ocean contains over 50 times more carbon dioxide (the main greenhouse gas) than the atmosphere,





with cold deep water serving as its main reservoir (Steward, 2008). Warming of the deep ocean layer could release large amounts of carbon dioxide into the atmosphere, intensifying the greenhouse effect.

In addition, thermal expansion of water (the steric effect) leads to increased sea levels; along with the loss of glacial mass, this accounts for approximately 75% of global sea level rise since the 1970s. From 1993 to
2010, the average global sea level rise was estimated as ~2.8 mm/yr, of which oceanic thermal expansion accounted for ~1.1 mm/yr (IPCC, 2013).

New, improved knowledge about the scale and drivers of changes in the physical properties of deep and intermediate water in the Nordic Seas is extremely important for a better understanding of global climate change and its recent intensification.

## 2. Data and methods

This study used hydrographic data from 1997 to 2016 obtained during annual cruises by the *R/V Oceania* in the Nordic Seas region. Each summer, the Institute of Oceanology, Polish Academy of Sciences (IO PAN) conducts
oceanographic measurements along about 15 sections, including approximately 200 CTD stations (Fig. 3) covering the area between northern Norway and the Fram Strait.

Oceanographic data from section N (Fig. 3) was used to analyze changes in deep and intermediate water properties. These data constitute the longest time series of observations collected by IO PAN in the Nordic Seas and are unique on an international scale, constituting a coherent data set: the same stations are always occupied
at the same time of year. Section N is situated along the 76° 30' N parallel, from 004 °E to 014 °E. Most of the section covers the warm and salty Atlantic Water (AW) inflow with two branches of the West Spitsbergen Current (WSC). The eastern WSC branch transports AW along the Barents Sea/Svalbard Shelf break, while the western WSC branch transports AW over the underwater ridge system (Walczowski, 2014). The AW occupies the upper layer of the water column to approximately 800 m depth. Arctic Waters occur west of the AW domain,
with the regions separated by the Arctic Front at ~007 °E.

In this study, the surface, intermediate and deep water layers were separated by depth. Water between 0 and 500 m was defined as surface water, between 500 and 1000 m as intermediate water and below 1000 m as deep water. This division corresponds to that previously defined by temperature and salinity (e.g., Schlichtholz and Houssais, 2002). For better comparison with published data, water properties at several levels were considered.
The mean values of the water mass properties were calculated for the whole region as well for the Arctic and Atlantic Water domains.

The CTD measurements were performed using an SBE 911plus probe and covered the whole water column from surface to seafloor. The measurement accuracy of temperature, conductivity and pressure was ± 0.001 °C, ± 0.0003 S/m and ± 0.015% of full scale, respectively, sufficient for analysis of slow rate changes in deep water.
Standard procedures provided by the manufacturer of the SeaBird system (software modules SeaSave and SBEDataProc) were used for receiving, processing and quality control of the hydrographic data.

Interpolation, quantitative data analysis and statistical calculations were performed in Matlab, while dedicated software (Ocean Data View) was used to visualize the processed data. Mean water properties and



water mass heat content were calculated from gridded fields. The data were interpolated using the Matlab griddata function, applying the linear interpolation method.

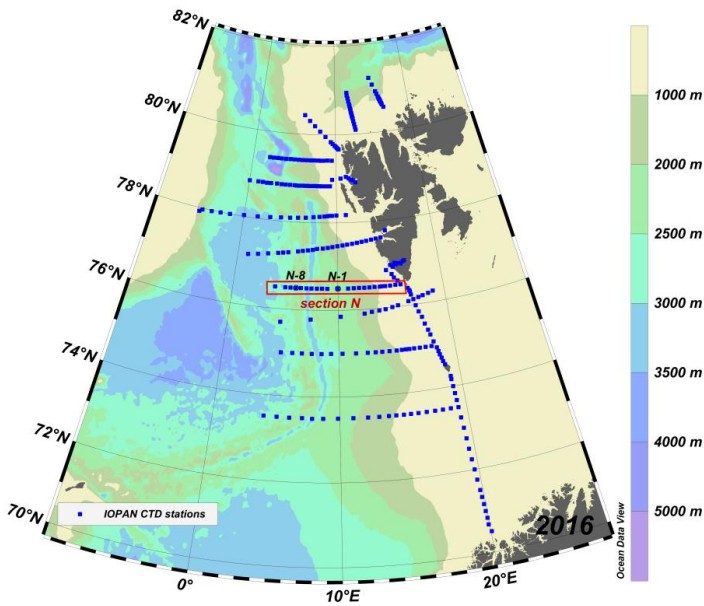

**Figure 3: Standard grid of CTD stations repeated annually during the AREX observational program carried by IO PAN. Red box encloses section N, used in this study.**

**3. Results**

The results show that, during the last 20 years, the deep and intermediate water in the studied region has warmed at least 10 times more (0.36 °C at 700 m) (Fig. 4) than the mean global ocean temperature change (0.03 °C at 700 m; IPCC, 2013). The average -0.80 °C isotherm depth was 1200 m in 1997 but 1700 m in 2016, 500 m

lower than 20 years earlier. The practical salinity of the water masses also changed considerably (+0.039 at 700 m) (Fig. 5). The highest increase in potential temperature and practical salinity was observed in the western part of the section, in the Arctic Domain (Figs. 6, 7).

After 2002, within all of section N, an average potential temperature of deep water below -1 °C was never recorded, nor was an average potential temperature of intermediate water below -0.7 °C (Fig. 4).

The dynamics of the water properties' temporal changes varied with depth (Figs. 4, 5). The potential temperature of intermediate water increased much faster (0.29–0.59 °C/20 yrs) than that of deep water (0.15–0.24 °C/20 yrs). The same pattern was observed for practical salinity of intermediate water (0.025–0.077/20 yrs) and deep water (0.0057–0.013) (Fig. 5). However, the increase in the potential temperature of the deep water was more stable, without major interannual variations.



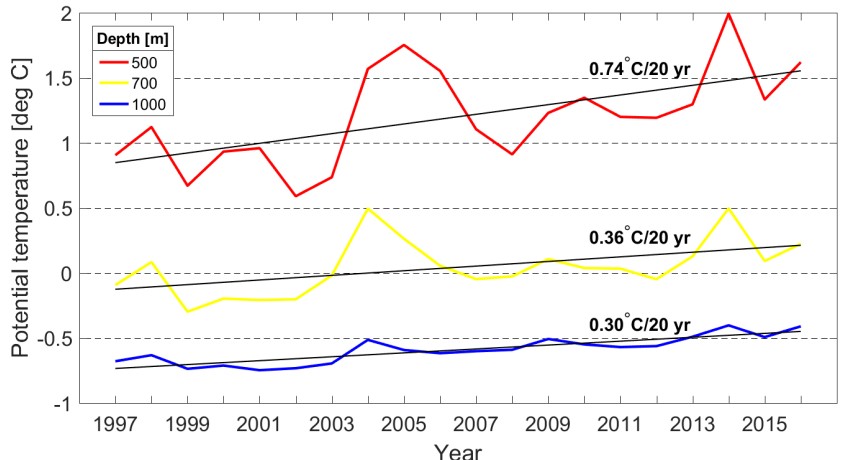

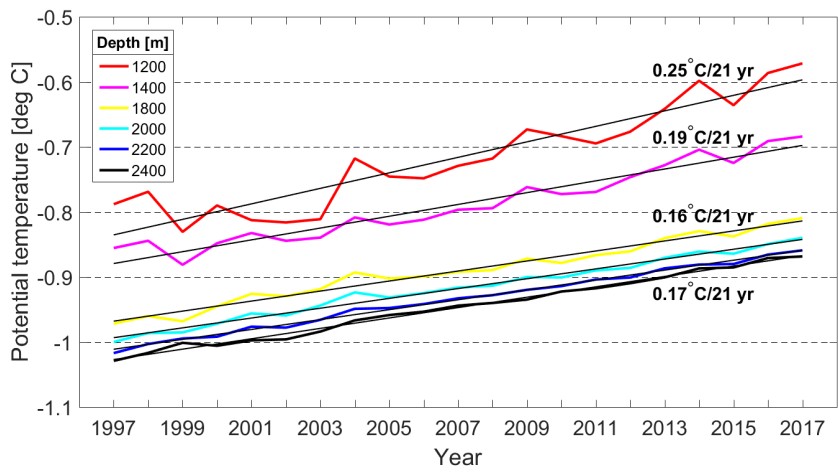

**Figure 4: Mean potential temperature of intermediate (upper) and deep (lower) water at selected levels across section N from 1997 to 2016. Note the differing temperature scales between the two graphs.**





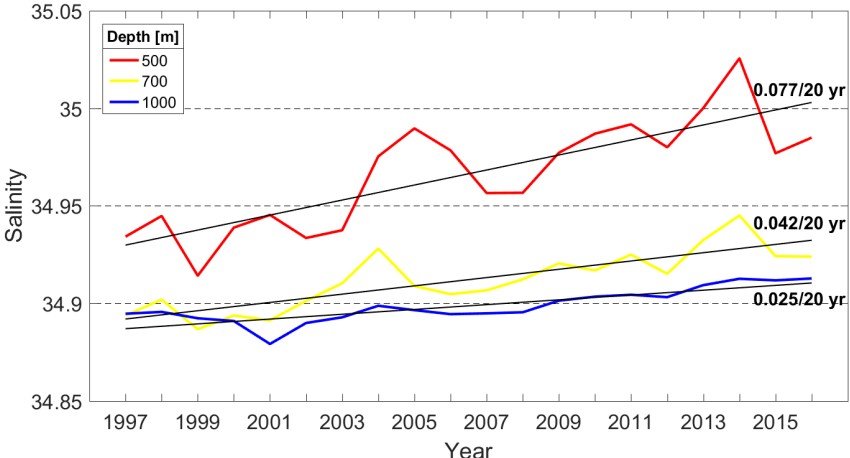

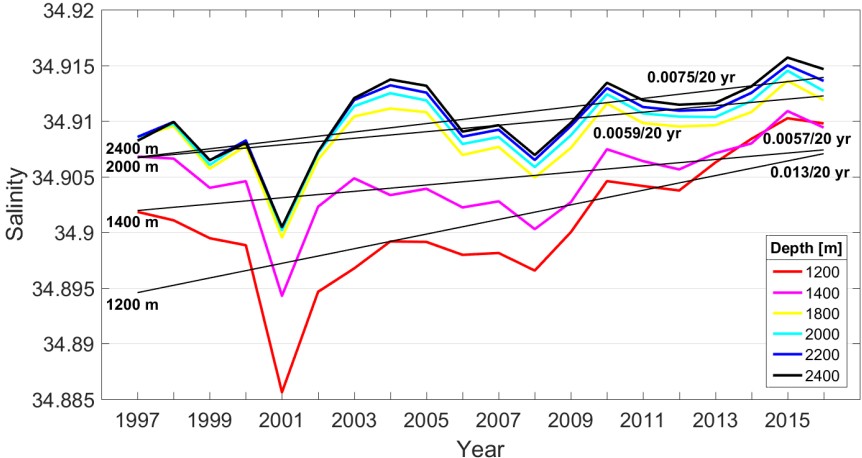

**Figure 5: Mean practical salinity of the intermediate (upper) and deep (lower) water across section N from 1997 to 2016.**





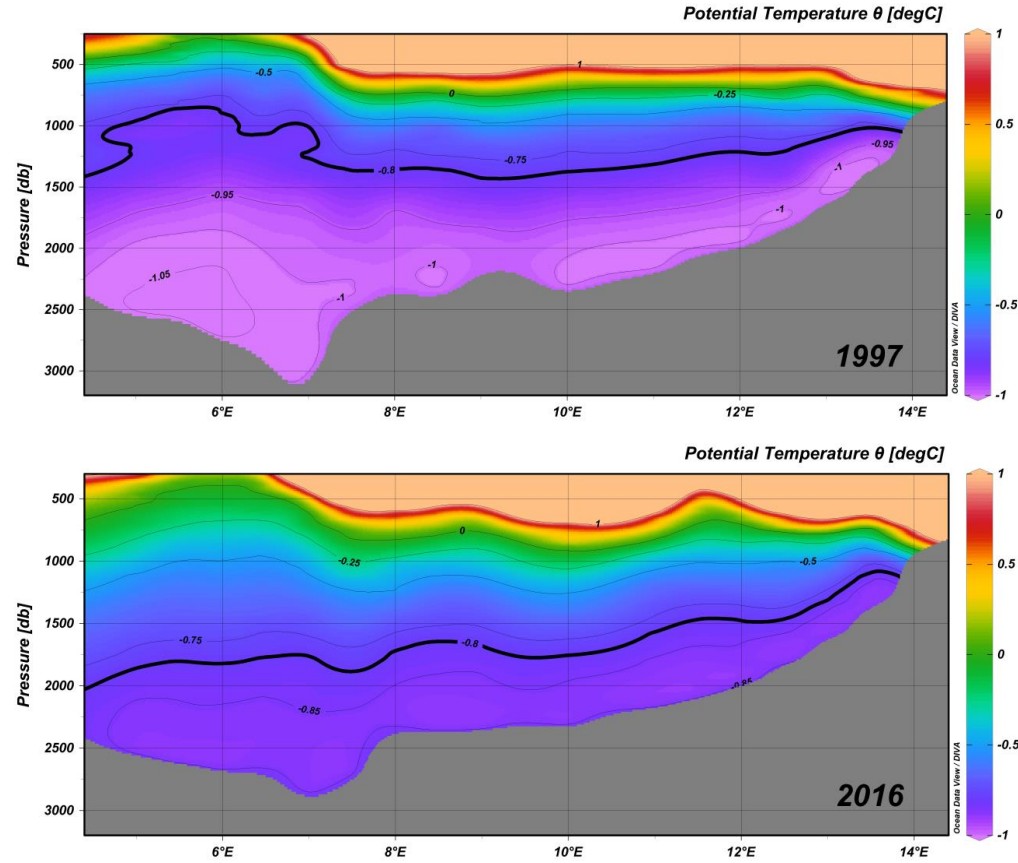

**Figure 6: Distribution of potential temperature along section N in summer 1997 (upper) and 2016 (lower). Bold line shows the -0.8° C isotherm.**

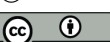



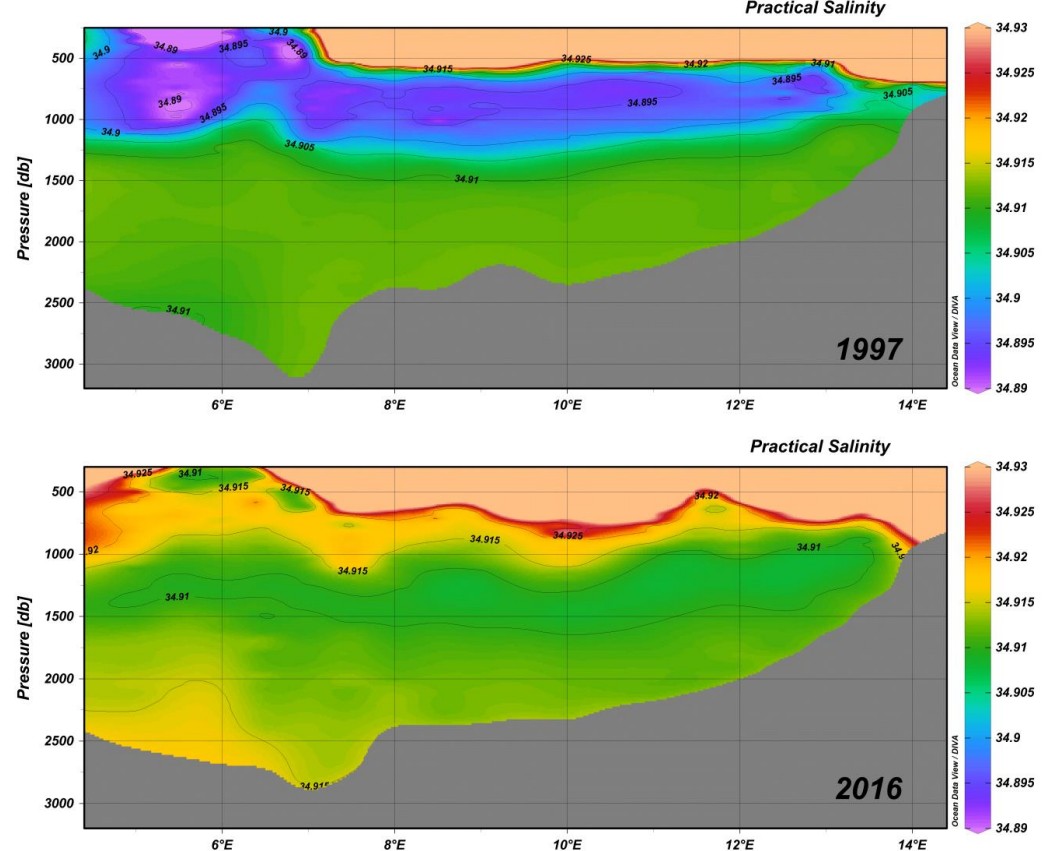

**Figure 7: Distribution of practical salinity along section N in summer**

**1997 (upper) and 2016 (lower).**

Over the last two decades, the heat content increased by 3978 MJ/m$^2$ in the entire water column of section N. This shows that deep and intermediate water are significant heat sinks in addition to surface water. The heat content of surface water rose by 1858.1 MJ/m$^2$ (47%), that of intermediate water by 838.5 MJ/m$^2$

(21%), and that of deep water 1281.2 MJ/m$^2$ (32%).

Fig. 8 shows the distribution of potential temperature at stations N-8 (76°30'N, 6°00'E) and N-1 (76°30'N, 10°00'E) (Fig. 3) from 1997 to 2016. The changes in potential temperature are different between these two stations because station N-8 is located in the Arctic Domain and station N-1 is in the Atlantic Domain.

At station N-8, the potential temperature of deep water did not exceed -0.95 °C after 2009, while the

potential temperature of intermediate water did not -0.8 °C. The lowest potential temperature of deep water in 1997 was -1.08 °C, while that in 2016 was -0.87 °C (Fig. 8). This means that the lowest potential temperature at this station increased by 0.21 °C in the last 20 years.

At station N-1 the potential temperature of the deep water did not exceed -0.95 °C after 2005, 4 years earlier than at station N-8, while the potential temperature of intermediate water did not exceed -0.7 °C. The



lowest potential temperature of deep water in 1997 was -1.01 °C, while that in 2016 was -0.87 °C (Fig. 8). This means that the lowest potential temperature at this station increased by 0.14 °C, in the last 20 years, 0.07 °C less than at station N-8.

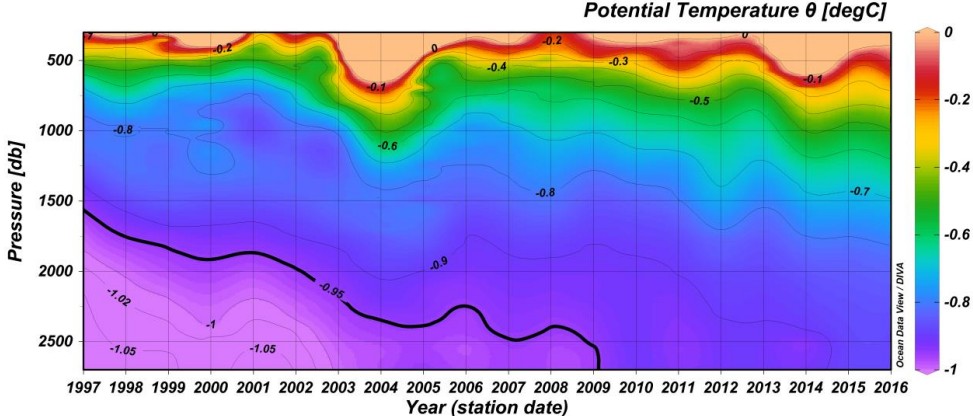

**Figure 8: Hovmöller plot of potential temperature at station N-8 (upper) and N-1 (lower) from 1997 to 2016. Bold line shows the -0.95° C isotherm.**

The Θ-S diagrams presented in Fig. 9 show the relationship between potential temperature and salinity for section N with regard to intermediate (500–1000 m) and deep (> 1000 m) water in 1997, 2007, and 2016.

The upper Θ-S diagrams show the significant influence of sinking Atlantic Water on the intermediate water column. In 1997, intermediate water (500–1000 m) was characterized by practical salinity below 35 and potential temperature of ~2 °C. In 2007, practical salinity was 35.05 and potential temperature reached 3 °C, while in 2016 practical salinity was 35.1 and potential temperature almost 4 °C.

The lower Θ-S diagrams show that in 1997, the deep water had varying properties in the western, central and eastern parts of section N. The western part was the least saline and coldest, the central part was the warmest and the eastern the most saline. In subsequent years, the deep water became more homogeneous, with its potential temperature and practical salinity constantly increasing.



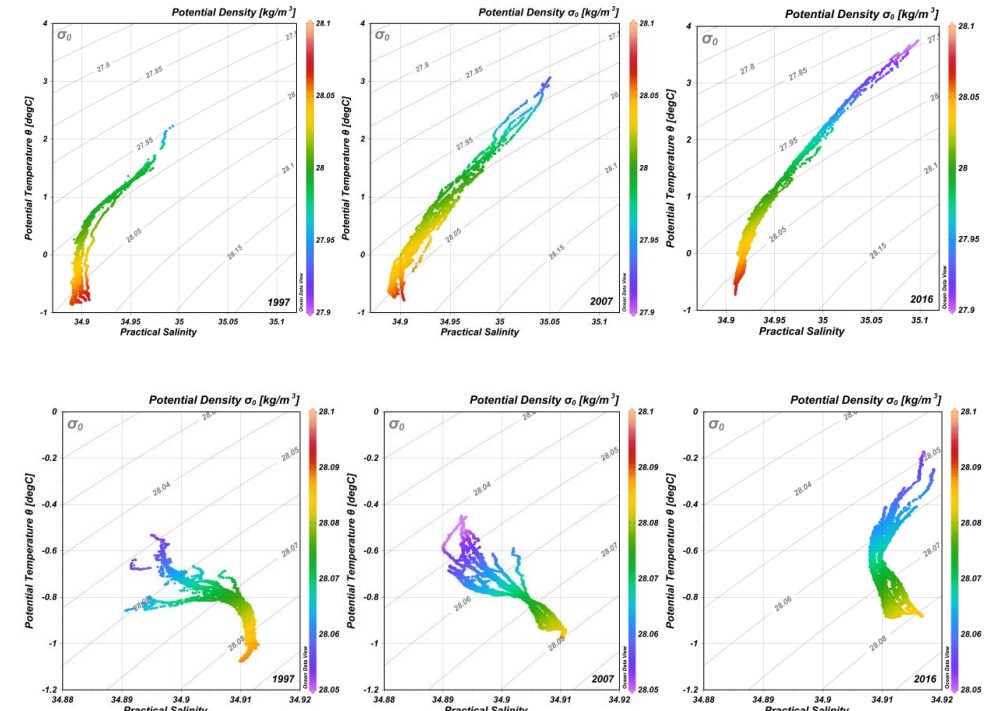

**Figure 9: Θ-S diagrams of intermediate (upper panels) and deep (lower panels) water for section N (Fig. 3) in 1997, 2007, and 2016.**

The increase in potential temperature and practical salinity of the deep water is most likely associated with at least two processes: (1) the weakening of deep convection in the Greenland Sea (Schlosser et al., 1991; Rhein, 1991), resulting in a reduced inflow of very cold, low-salinity water and (2) the intensified inflow of deep water from the Arctic Ocean into the Nordic Seas; this is warmer and more saline than deep water from the Greenland Sea (Rudels, 1986; Rudels et al., 2012).

The impact on intermediate water includes warmer, more saline Atlantic Water sinking during winter cooling. The potential temperature increase of deep water is less substantial than for intermediate water but is more stable, without interannual fluctuations.

## 4. Summary

In the last two decades, deep and intermediate water in the study area has warmed at least ten times more (0.36 °C at 700 m) than mean global values (0.03 °C at 700 m). The increase in the water potential temperature, especially strong in the western part of the section, was associated with considerable changes in the water masses' practical salinity. With regard to deep water, these changes were most likely affected by the weakening





of deep convection in the Greenland Sea. This, in turn, may be the effect of a reduced inflow of very cold, low-salinity water, as well as the intensified inflow of deep water from the Arctic Ocean into the Nordic Seas, as this is warmer and more saline than deep water from the Greenland Sea. In comparison, intermediate water was primarily impacted by relatively warm and saline Atlantic Water, which sank through water column due to heat loss. In contrast to the variability in deep water physical properties, the temperature increase of intermediate water was more substantial with considerably higher interannual fluctuations.

A mean heat content increase of 3978 MJ/m$^2$ per 20 years requires an additional 6.3 W/m$^2$ of atmosphere-ocean heat flux, almost 3 times more than greenhouse effect mean radiative forcing. Although the study region is an effective heat sink, surplus of this heat stored by ocean and transported by the WSC to the Arctic Ocean may have an increased influence on Arctic sea ice melting and climate change.

Changes in the deep and intermediate water properties in the area may have much stronger impacts on the recent intensification of climate change than previously forecast, increasing ocean circulation, the amount of dissolved oxygen and carbon dioxide in the deep sea, and sea level rise at a much faster rate.

*Competing interests.* The authors declare that they have no conflict of interest.

*Acknowledgements.* This study was funded by the National Science Centre, Poland within the DWINS Project (2016/21/N/ST10/02920). We would like to thank the crew of *R/V Oceania* for their support and help at sea.

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
