# Peer review of "Changes in the properties of deep and intermediate water masses in the Nordic Seas from 1997 to 2016"

_Ocean Science, 2018_

## Referee Comment (RC1) · Anonymous Referee #1 · 18 Sep 2018

This manuscript presents observations of temperature and salinity over 20 years along a section in the southwestern Fram Strait. The authors show the increase in temperature and salinity below 500 m. Besides this, the paper does not present further scientific results or discussion and I am very much wondering if this justifies publication in OS. Furthermore, even the limited content is poorly presented. Hence I do not recommend publication. However, since the authors show in Fig. 3 the existence of a very valuable larger data set, I highly recommend extending the ms. to the full data set, substantially revise text and figures, and submit it to Earth System Science Data.

The ms. refers to the extensive temperature and salinity data set obtained by the Polish

[Figure]

IO PAN, but it does not present these data (at least it seems so, but I am not sure, see below). From the single section that is presented, the authors show an increase of both parameters, but other than stated in the abstract and the summary, no further analysis on causes or impacts are given. Raise in temperature of deep Fram Strait/Greenland Sea temperatures and salinities have been shown before and analyses of causes have been discussed for example by Langehaug et al., 2012, Somavilla et al., 2013, von Appen et al., 2015. None of these papers or others on this topic are referred to. (The paper by Langehaug et al., is referred to, but only because the authors use their circulation sketch! Apart from that, they do not even mention this extensive study on exactly the same topic!) In the abstract and the summary, the authors make some statements (but no analysis) that are either very general (global ocean warming) or highly speculative. It is unclear why the authors do not use more of their own data, not to speak of other available data, to put their findings into a context. The ms. contains a lot of unnecessary information; the text has several repetitions; it is often imprecise, and can be shortened substantially. On the other hand, important information is missing. Hence, even before submitting anything to ESSD, the ms. should be revised substantially.

A few detailed comments might illustrate the problems:

Title: No water masses are analyzed.

Abstract: The 1st paragraph is repetitive and can be summarized in one sentence. It promises analysis of potential drivers, but the ms does not contain such analysis. The second paragraph can be skipped entirely – it speaks about the impact of the findings but no in-depth consideration is given in the ms.

The introduction largely reads like a working report of the IO PAN (1st paragraph), or like a justification for the funding request for further ocean observations in general. The second para speaks about recent studies without citing them. The last sentence of para 3 (p 2 line 19) is a bizarre statement – how would intermediate layers absorb heat from the surface layers (surface is here, by the way, defined as reaching down to

500 m!)? No arguments or explanation for such a process are given later. Fig. 2 can be skipped. Page 3, Line 10: This statement is again bizarre. Purkey and Johnson (2010), a key paper on ocean warming (that the authors do not refer to), or Desbruyeres et al, 2016, use high quality data after 2005 in all oceans.

Data and methods: Fig. 3: Most of the shown data are not used. Why are they shown? Page 4, line 13ff says that data from 1997 until 2016 are used. Then the authors mention 15 sections that obviously are not used. Para 2 says that section N (the one which finally IS used) provides the longest time series. What does that mean? Is that longer than the 20 years mentioned above? Why do the authors not give the simple information about the data used (time and location) in one clear sentence? Line 20ff speaks about water masses (AW) and branches – what does this explain in the chapter "data or methods"? On the other hand, neither here nor later in the ms., any information is given on the definition of water masses or where any branches can be seen in the data or why they are of importance here. So what shall we do with this information? Line 26ff) introduces three layers, although the title and the results deal only with two layers. In line 23, the authors write that AW extends to 800 m. Hence it forms part of the so-called "intermediate layer" between 500 and 1000 m. Hence there must be another water mass contributing to the "intermediate layer". No information is given about that one. The authors should make up their mind whether they want to deal with water masses, yes or no.

Take just the one sentence (page 4, line 30) "The mean values of the water mass properties were calculated for the whole region as well for the Arctic and Atlantic Water domains." At least four questions arise: 1. What are Atlantic and Arctic domains? 2. Which water masses are the authors talking about? 3. How could they possibly compute a mean value of water masses if the layers contain fractions of several water masses (see above)? 4. What means the "whole region"? Is that the region covered by the whole data set presented in Fig. 3?

I do stop here, since I hope enough comments are given on where the problems are

and what should be done to improve the ms. for a possible submission to ESSD.

---

## Referee Comment (RC2) · Anonymous Referee #2 · 18 Sep 2018

General comments

The authors presented an analysis of a relatively long measurement time series of salinity and temperature transects in the northern North Atlantic. They presented linear regression analyses, transects and Hovmöller plots to examine the changes in the CTD measurements. The strength of this study is that they focus on indermediate and deep layers of this region, where only few continuous measurements exist. However, there are substantial aspects to improve until it is ready for publication. This affects among others the literature research, statistical significance analysis, text structure and discussion of the results. Moreover, they showed the full amount of the dataset available

which could be used to proof their conclusions, which are merely assumptions.

Specific comments

Title It says "Nordic Seas" which includes Iceland, Norwegian and Greenland seas, but you used only one transect (according to the map in Figure 1) close to Fram Strait. Thus, you cannot say that you analysed all of the adjacent seas. Additionally the word "properties" seems too general. Maybe you should change it to something like: "Changes in temperature and salinity in a transect close to the Fram strait from 1997-2016 and their impact on the Nordic Seas" in order to get the main point of the manuscript as is. However, if you change it substantially the title may have tob e changed anyway.

The Abstract is too general. What is new? 'may' should be prohibited, you could say: "Our research indicates that..."

Introduction The authors used very few references and described the state of the art very poorly. What is the importance of the arctic seas in the global climate? How do they impact sea ice? Please use more references and describe recents findings more intense.

Methodology You only use data from your institute. Why? Are these the only measurements. Is it due to better consistency? You should explain it once. This way it seems that only your institutes has such data, which is definately not true. Have these data been used before? For what purpose and what are the results (should be part of the more detailed Introduction). You analysed the trends without any significance tests which should definately be included.

Results The figures and results sound more like an overview of the data than a scientific analysis. Moreover, the presentation of the results are very confused and do not seem to have a clear thread. Your first conclusion in the summary should be your research question for a new submission. You have the data (as it seems from Figure 3) to show

how the flow of water masses through the Fram Strait have changed, if you use more than one transect.

All in all, in order to make this work worth for publication, the authors should dig much deeper, examine statistical significances and use appropriate figures to show their results. Detailed examples are shown below.

Technical comments

Page 1 l. 9-10: sentence is incomplete. "was/is paid"

l. 11: The period must not be mentioned again

l. 13: "salinity has also changed" How? Why? Write more than a half sentence.

Page 2 Figure 2 from IPCC report can be explained in the text with a direct reference to this picture in the IPCC report. The figure itself can be skipped.

l.6: sounds like, only your institute works with this topic. Generally, it reads like a praise to your institute, which is not scientfic.

Page 4 l. 17: Specifiy that section N is located west of Spitzbergen. Thus it is easier for the reader to follow. Why particularly this transect and not another? Can't you compare different transects?

l.20: which time of the year? Important for discussion and comparison with global rates. (You have written "summer" in l. 14, but you should also specify the months. Are the global values you compared your results with also from summer?)

Page 5 l.11 The authors should start describing the figure briefly before analysing them. Why did you choose potential temperature? What is the difference analysing absolute and potential temperature and what are the advantages/disadvantages (you should put that in the methodology part). Generally you jump from figure to figure without a clear thread.

Subtitle of Fig.4/5: Trends and their unit (per decade, per year?) are missing. What means "across section N"? Is it a mean value? Are the changes in absolute changes during the 21/20 years? For better comparison it should be converted to K/psu per decade.

Page 9 How did you estimate the heat content, if you only have a transect? Where are the results shown? First, the comparison of heat content changes in the surface and the deeper layers should been shown and then the discussion about the importance can be done. However, the same results could also be drawn from the temperature changes. Moreover, the total heat content is not comparable due to the different layer depths and the bathymetry. Relative changes should be used.

Page 10 Figure 8: Why do you have pressure on y-axis? In the text you always refer to depth in m. The readability would be improved if it were consistent in such cases.

Page 11 Figure 9: The colourbar should be mentioned in the caption, because it is not so clear from the figure itself. Why do you show these theta-S diagrams? Minima, Maxima and changes could also be seen in the figures before, thus the strengt hof theta-s-diagrams has not been used. (What does it mean that the direction of the points is flipped by approximately 90° between intermediate and deep layer? And that it seems to flip in 2017 in the deeper layer?)

l. 10ff.: How do you know that 1) and 2) is happening, it is just an assumption. You could (and should) prove this with the other cross sections shown in Figure 3!

l. 20ff Is the global mean value annually or from summer? It may not be comparable with your observations during summer. Please discuss it.

Page 12 l.7ff:This is a new results and should be part of the discussion of the results. It does not belong to the summary or the conclusions. "May have increased influence on Arctic Sea ice melting and climate change" is very simple and sketchy.

---

## Author Comment (AC1) · 12 Oct 2018

We would like to thank the Anonymous Referee #1 for the insightful and constructive review. It will certainly help us improve the quality of our papers in the future. The main Referee objection concerns length and structure of the paper, as well as the lack of citations. Additionally, reviewer list papers which should be cited. Our aim was writing a short paper with a simple message. In our opinion, readers are tired of reading long introductions with a tenth of citations. The short, synthetic paper may be also interesting and valuable. Especially when the message concerns years of measurements. Of course, we are aware that a common system of scientist's evaluation forces putting a

lot of citation. Langehaug et al., 2012, Somavilla et al., 2013, von Appen et al., 2015 show processes in the Nordic Seas in a different context, besides, the data they present also needs updating. However, we agree with Referee # 1 that paper needs improving. After this general statement, we answer specific comments of the Anonymous Referee #1. Ref. #1. This manuscript presents observations of temperature and salinity over 20 years along a section in the southwestern Fram Strait. Ans. Our idea was rather to present temperature and salinity observations upstream of the Fram Strait. The section along the 76° 30' N parallel should not be treated as the Fram Strait, not even the south Fram Strait. Ref. #1. The authors show the increase in temperature and salinity below 500 m. Ans. We also show an increase in temperature at other levels. Ref. #1 the paper does not present further scientific results or discussion Ans. The discussion will be improved. Ref #1. . . . the existence of a very valuable larger data set, I highly recommend extending the ms. to the full data set, substantially revise text and figures, and submit it to Earth System Science Data Ans. Yes, it is a good idea. But here we want to show part of this data set with a focus to the specific phenomenon. Ref #1 Raise in temperature of deep Fram Strait/Greenland Sea temperatures and salinities have been shown before and analyses of causes have been discussed for example by Langehaug et al., 2012, Somavilla et al., 2013, von Appen et al., 2015. Ans. Langehaug et al. (2012), analysed the section located in the northern Fram Strait. Also von Appen et al., 2015 described mostly the northern part of the Fram Strait. Our experience shows that this is the most unstable region, with different hydrological conditions than in the area we analysed. From the other hand, Somavilla et al., 2013, shows north Atlantic to 005°E. Nevertheless, we will improve our article and cite these (and other) papers. Ref #1. It is unclear why the authors do not use more of their own data, not to speak of other available data, to put their findings into a context. Ans. We agree, that IOPAN has a lot of data. We chose the 'N' section along the 76° 30' N because it is our longest time series and is located in an interesting and important region. We used data only from our Institute because we wanted them to be as consistent as possible and performed exactly in the same period of time, from June 22 to July 22, each of the

20 presented years. In addition, Walczowski (2014) showed that the salinity and temperature time series at this section may be representative for the larger Nordic Seas region. Ref. #1The ms. contains a lot of unnecessary information; the text has several repetitions; it is often imprecise, and can be shortened substantially. Ans. We made the text as short as possible, but we will try to be more precise and avoid repetition. Ref #1. Title: No water masses are analysed Ans. Yes, it is imprecise and title will be changed Rev#1. Abstract: The 1st paragraph is repetitive and can be summarized in one sentence. . . . Ans. We made abstract short intentionally, improving paper we will do it more essential. Ref #1. The second para speaks about recent studies without citing them. Ans. We cite IPCC and Levitus. But we will cite much more. Ref #1. The last sentence of para 3 (p 2 line 19) is a bizarre statement – how would intermediate layers absorb heat from the surface layers Ans. In the short time scale the geothermal heating of ocean floor is really small (50-200 mW/m2, Sclater et al., 1980; Kadko and Baross, 1995; Stein et al., 1995; Murton et al., 1999). According to Hofmann and Maqueda, 2009 the mean GHF is 100 mW/m2, therefore in the global scale, the heat in the intermediate and deep waters may be gained only from surface ocean layer. Here we do not consider the processes of the heat transporting from surface to deep ocean, only results. Ref #1 surface is here, by the way, defined as reaching down to 500 m Ans. Yes, better will be 'upper layer', 'upper ocean'. Ref#1. Fig. 2 can be skipped. Ans. Fig. 2 is very important, but we will leave only results in descriptions and reference to the IPCC report. Ref. #1 Page 3, Line 10: This statement is again bizarre. Purkey and Johnson (2010), a key paper on ocean warming (that the authors do not refer to), or Desbruyeres et al, 2016, use high-quality data after 2005 in all oceans. Lack of good quality data is mentioned in the IPCC report as well as by Levitus (2012). Purkey and Johnson (2010) and Desbruyeres et al. (2016) analyzed data after 2005 in all oceans, however, not in the Nordic Seas and Arctic Ocean (Fig. 1: Desbruyeres et al. (2016), Fig.8: Purkey and Johnson (2010). "Due to a lack of deep repeat hydrographic data, the Arctic Ocean is not included in the analysis." Desbruyeres et al. (2016) Ref #1 Fig. 3: Most of the shown data are not used. Why are they shown? Ans. To show that our

measurements are not incidental and cover bigger region Ref #1. Page 4, line 13ff says that data from 1997 until 2016 are used. Then the authors mention 15 sections that obviously are not used. Para 2 says that section N (the one which finally IS used) provides the longest time series. What does that mean? Is that longer than the 20 years mentioned above? Period 1997-2016 covers 20 mentioned years. Figure 3 presents 'standard' grid of stations covered by us since 2000. Measurements at section 'N' were performed earlier and therefore it is our longest time series. Ref #1. Line 20ff speaks about water masses (AW) and branches – what does this explain in the chapter "data or methods"? Ans. We describe a region of measurements, the 'data and methods' chapter seems to be good for this. Ref #1. any information is given on the definition of water masses or where any branches can be seen in the data or why they are of importance here. Ans. We do not analyse water masses (as we describe later), but again, in our opinion, the general information about hydrographic conditions should be provided. The information about the location of both branches is given (eastern over the shelf break, western over the underwater ridge system). Additionally, the position of the Arctic Front is given (007° E). Ref #1. Line 26ff) introduces three layers, although the title and the results deal only with two layers. Ans. Indeed the main focus is on the deep and intermediate layers. Should we do not write any word about the upper layer? Ref #1. What are the Atlantic and Arctic domains Ans. Arctic Waters occur west of the AW domain, with the regions separated by the Arctic Front at ∼007 °E. (p 4, paragraph 25) Ref #1. Which water masses are the authors talking about? Ans. Yes, we agree, the 'water masses' will be changed for simple 'water' Ref#1 What means the "whole region"? Ans. It is the whole section. Will be improved.

---

## Author Comment (AC2) · 13 Oct 2018

Review response #2

We would like to thank the Anonymous Referee # 2 for the constructive and inspiring review. It will help us in improving this paper and future works.

We agree that there is no statistical significance analysis in our work, which we are going to complete. The literature research will be improved, as well as the results discussion. We consider using data from section upstream and downstream of section 'N' for better presenting our results.

Specific comments

*Ref#2. Title It says "Nordic Seas" which includes Iceland, Norwegian and Greenland seas, but you used only one transect (according to the map in Figure 1) close to Fram Strait.*

Ans. Yes, we agree that the title is too general. It will be changed. The proposition given by Referee#2 is very good. Also abstract will be improved. In introduction, we will describe climatic importance of the Nordic Seas more widely, with more citations.

*Ref#2 Methodology You only use data from your institute. Why? Are these the only measurements. Is it due to better consistency?*

Ans. We decided to use only data from our institute, due to the data consistency. There are data collected in the same points, in the same year period, processed in the same way. It will be explained better.

*Ref#2 You analysed the trends without any significance tests which should definitely be included.*

Ans. We will include the significance analysis. All presented trends are statistically significant, with p-values less than 0.01.

We have a lot of data, however, due to the specificity of the Nordic Seas region, often due to very bad weather conditions, we are not able to complete all stations. We chose the N section because it is our longest time series and is located in an interesting hydrographically place where dense water flows both from the central part of the Greenland Sea and from Storfjord.

*Ref#2 Which time of the year? Important for discussion and comparison with global rates.*

Ans. We performed measurements exactly in the same period of time, from June 22 to July 22, each of the 20 presented years. We agree that it is very important for comparison with global rates.

*Ref#2 Trends and their unit (per decade, per year?) are missing. For better comparison it should be converted to K/psu per decade.*

Ans. Trends are calculated per two decades. In Figure 4 (lower) trends per 21 years are wrong. This is our oversight. You are absolutely right that we should use trends per decade for better comparison.

*Ref#2 Why do you have pressure on y-axis? In text you always refer to depth in m.*

Ans. We agree that we should use one unit in the article for better readability.

*Ref#2 This is a new results and should be part of the discussion of the results.*

Ans. You are right that it should not be in the summary or discussion, but in results.

Of course, we are going to improve all the oversights and introduce necessary changes.